# MTHFR SNPs (Methyl Tetrahydrofolate Reductase, Single Nucleotide Polymorphisms) C677T and A1298C Prevalence and Serum Homocysteine Levels in >2100 Hypofertile Caucasian Male Patients

**DOI:** 10.3390/biom12081086

**Published:** 2022-08-07

**Authors:** Arthur Clément, Edouard Amar, Charles Brami, Patrice Clément, Silvia Alvarez, Laetitia Jacquesson-Fournols, Céline Davy, Marc Lalau-Keraly, Yves Menezo

**Affiliations:** 1Laboratoire CLEMENT, 75016 Paris, France; 2American Hospital, 92200 Paris, France; 3Cabinet Medical, 15 Avenue Poincarré, 75016 Paris, France; 4Cabinet Médical Endocrinologie, 40 boulevard de Courcelles, 75017 Paris, France; 5Cabinet Médical, 34 Avenue d’Eylau, 75016 Paris, France

**Keywords:** hypofertility, MTHFR SNP, C677T, A1298C, homocysteine, Caucasian men population

## Abstract

Methylation is a crucially important ubiquitous biochemical process, which covalently adds methyl groups to a variety of molecular targets. It is the key regulatory process that determines the acquisition of imprinting and epigenetic marks during gametogenesis. Methylation processes are dependent upon two metabolic cycles, the folates and the one-carbon cycles. The activity of these two cycles is compromised by single nucleotide polymorphisms (SNPs) in the gene encoding the Methylenetetrahydrofolate reductase (MTHFR) enzyme. These SNPs affect spermatogenesis and oocyte maturation, creating cytologic/chromosomal anomalies. The two main MTHFR SNP variants C677T (c.6777C>T) and A1298C (c.1298A>C) together with serum homocysteine levels were tested in men with >3 years’ duration of infertility who had failed several ART attempts with the same partner. These patients are often classified as having “idiopathic infertility”. We observed that the genetic status with highest prevalence in this group is the heterozygous C677T, followed by the combined heterozygous C677T/A1298C, and then A1298C; these three variants represent 65% of our population. Only 13.1% of the patients tested are wild type (WT), C677C/A1298A). The homozygous 677TT and the combined heterozygote 677CT/1298AC groups have the highest percentage of patients with an elevated circulating homocysteine level of >15 µMolar (57.8% and 18.8%, respectively, which is highly significant for both). Elevated homocysteine is known to be detrimental to spermatogenesis, and the population with this parameter is not marginal. In conclusion, determination of these two SNPs and serum homocysteine should not be overlooked for patients with severe infertility of long duration, including those with repeated miscarriages. Patients must also be informed about pleiotropic medical implications relevant to their own health, as well as to the health of future children.

## 1. Introduction

Methylation is a ubiquitous biochemical process that covalently adds methyl groups to several types of molecules. It contributes to the tertiary structure of lipid membranes and transport/receptor systems and is involved in the synthesis of various hormones and biogenic amines and completes brain maturation. Via methylation of proteins (mainly histones) and DNA, it plays a critical role in two major regulatory and developmental processes during reproduction: imprinting and epigenesis [1,2,3,4]. Methylation determines the complementary regulatory characteristics of male and female genomes [1,2]. Methylation processes rely on two interconnected metabolic cycles: the folates (FolC) and the one carbon cycle (1-CC), (Figure 1). Post-methylation by the universal methylation co-factor SAM (S adenosylmethionine), SAH, S-adenosyl homocysteine, and then homocysteine (Hcy) are released. Both compounds are inhibitors of methylation processes. [5]

The 1-CC allows recycling of homocysteine (Hcy) to methionine (Met). The 1-CC is supported through associated metabolic pathways, but the FolC is crucial; this supplies a methyl group for 5 methyltetrahydrofolate (MTHF) formation via methionine synthase (MS). This formation of 5MTHF encounters a critical step at the level of the MTHFR enzyme, methylene tetrahydrofolate reductase, which converts 5-methylene tetrahydrofolate to MTHF. The MTHFR enzyme is subject to single nucleotide polymorphisms (SNPs) that affect its activity [6,7]. The two most significant SNPs are C677T(Ala222Val) and A1298C(Glu429Ala); they impair MTHFR activity resulting in defective Hcy recycling, characterized by an increase in circulating Hcy, a cause and consequence of oxidative stress (OS) [8]. OS is a well-known hazard in the methylation process [9,10]. Carriers of these 2 MTHFR isoforms may reduce their methyl folate production up to 17% for MTHFR A1298C and 70% for MTHFR C677T. The resulting folate deficiency jeopardizes DNA stability and methylation processes [11]. The trilogy MTHFR SNP, Hcy and folate deficiency have a major impact on male [12,13,14,15,16,17,18,19,20] and female [21,22,23,24,25] gametogenesis, embryogenesis implantation/miscarriages [26,27,28], and even pregnancy complications “from gametes to infant delivery”. Associations between maternal, paternal, and then fetal MTHFR gene C677T and A1298C polymorphisms will affect DNA stability [21].

The maternal impact on embryo development is always highlighted and emphasized in relation to neural tube defects and the other obstetrical complications; the impact of paternal MTHFR SNP is generally overlooked. However, a correct methylation process facilitates correct timing of the first embryo cleavages and blastocyst formation [29]. As mentioned earlier, [12] homocysteine has a negative impact, and DNA methylation anomalies as a negative factor in sperm quality is now well documented [12,13,14,30,31]. All of our patients with a history of idiopathic infertility of long duration, two miscarriages or two failed IVF/ICSI attempts, and with no specific shifts in classical fertility parameters in either partner, are now tested for these two isoforms. We present here the epidemiologic distribution of these SNPs and the resulting impact on circulating homocysteine in 2127 men.

## 2. Materials and Methods

### Population and Ethical Considerations

Determination of the two SNPs and serum homocysteine levels has been a standard procedure in our infertility units since 2019. In our practice, these tests are classically recommended for patients suffering infertility of at least two years’ duration with two failed IVF/ICSI attempts and/or at least two miscarriages. The test is not performed in patients with nonobstructive azoospermia. They are submitted to the classical ethics regulations recommended by the French “Agence de Biomedecine”. Ethical committee approval is not required, but testing must be prescribed by a certified (by agence Regional de Santé: Regional Health Agency) andrologist, gynecologist/obstetrician, or endocrinologist, for patients seeking fertility treatments in authorized/approved clinics or hospitals. Signed informed consent is mandatory; testing for the purpose of building a control group in a fertile population is not permitted. Patient informed consent is also required to allow anonymous publication of the data. The analyses must be performed in certified/licensed laboratories and are not reimbursed by social security. From February 2019 to December 2021, 2439 women and 2127 men were tested. Some of the patients declined testing.

*Serum Homocysteine:* The protocol has been previously described [32]. *Briefly:* Fasting blood samples were collected in the morning, and serum Hcy measured using the VYTROS kit, which allows determination of homocystine and homocysteine. Homocystine is reduced to homocysteine with tris(2-carboxyethyl), and total homocysteine is then transformed into cystathionine in the presence of cystathionine beta synthase (CBS). Cystathionine is hydrolyzed by cystathionine lyase to form Hcy, ammonia, and pyruvate. Pyruvate is reduced to form lactate; the amount of NAD+ produced by lactic acid dehydrogenase is proportional to all the homocysteine present in the sample, and this is measured by spectrophotometry at 340 nm. The assay is linear from 1 to 90 μM-homocysteine.

*MTHFR SNPs:* The LAMP (loop isothermal amplification) human MTHFR mutation kit based on a hybridization technique was employed, which requires a 5-μL blood sample. Amplification is performed at 65 °C, using several sets of primers simultaneously. Six specific primers covering the locus of the mutation are used for the C677T SNP. The same protocol was used for A1298C SNP, with 6 specific primers covering the region of the mutation. Two loop primers are used in both, and the probes simultaneously amplify the wild type gene. The results were evaluated by comparing the curves obtained by fluorescence (Figure 2). The full protocol has been previously described [33].

## 3. Results

### 3.1. Distribution of the Two SNP Combinations

The genetic status with highest prevalence is C677T, followed by the combined heterozygote C677T/A1298C, and then A1298C; these three variants represent 65% of our population. Only 13% of our patients tested as wild type (WT), C677C/A1298A. (Table 1).

### 3.2. Impact of the SNP Combinations on Hcy Levels >15 µMolar

The percentage of patients showing elevated serum Hcy (>15 µMolar) is strongly dependent on the type of MTHFR SNP (Table 2, Figure 3). Variant combinations with the greatest impact are T677T, the combined heterozygous mutation C677T/A1298C, and heterozygous C677T. Homozygous T677T/A1298A and the combination of C677T/A1298C significantly increase serum Hcy levels compared to the WT (57.8% and 18.8%, respectively, vs. 8.6%). No significant difference is observed for the other combinations. Compared to WT, the Hcy level is elevated by 14.1× in the homozygous 677TT group and by 2.5× in the combined 677CT/1298AC group.

In total 387 patients were found to have Hcy >15 µMolar (387/2127 = 18.2%).

## 4. Discussion and Conclusions

The presence of MTHFR variants and homocysteine levels are parameters that are usually neglected in physiopathology. However, these are not benign and are a source of several pathologies [34,35,36]. They do affect fertility [13,14,15,16,17,19,21,28,37]. These isoforms are highly prevalent in Mediterranean and Latino populations and in Chinese Han and Zhuang ethnicities, where it can reach 70% of the population. Only 13.1% of the male patients and 11% of the 2439 women tested are completely free of any mutation. This is much lower than is observed in two fertile populations (between 50% and 60% [38,39]). The most hazardous combination is the homozygous T677T. More than one half (57.8%) of affected patients have an elevated Hcy; its prevalence is 13.1% in our overall male population and 11% in the women tested. This is much higher than has been described in fertile populations (between 6 and 9% [38,39]). The combined C677T/A1298C heterozygous variant population is 21.4% in males and 19.6% in the 2439 women tested; this also presents a hazard, as it significantly increases the risk of elevated serum Hcy (18.8% of the population). The prevalence of this double heterozygosity is between double and triple of the prevalence observed in fertile male parents [38,39]. Globally, 18.2% of our male population has an elevated circulating homocysteine; only 82/2439 = 3.4% of the women tested have an elevated homocysteine. This is essentially due to an estrogen-dependent capacity to activate the cystathionine beta synthase pathway [40] that eliminates homocysteine and releases cysteine but does not solve the problem of methylation. In the wild type group, 8.6% of the male population shows elevated homocysteine; this represents 1.1% of the total population tested. A very small portion (8/2127) of our population has a triple mutation; this type of combination has been previously observed [7]. Clearly, a quadruple mutation must be lethal. The consequences of these MTHFR SNPs on male gametes have been reported in several meta-analyses [17] all over the world. Recent observations have emphasized the negative impact of the resulting hypomethylation on decline in fertility [30,31] and on the decreased developmental capacity of the early embryos carrying these variants [21].

Homocysteine also has a major negative impact as a cause and a consequence of oxidative stress [8,11], an important issue in infertility. The Rotterdam Periconception cohort has recently recommended “*routine analysis of homocysteine levels in preconceptional and pregnant women and their partners*” [41] in order to avoid the negative impact of homocysteine imbalance on gametogenesis and early embryo development. Therefore, both isoforms must be tested, although 1298AC appears to be of lesser interest. Determining a critical level of Hcy for health is also a matter for debate. The pathological impact of elevated serum Hcy and methylation defects is recognized in cardiology, psychiatry, and cancer and in obstetrics in relation to neural tube defects. The latter problem has led to national fortification programs that supplement foodstuffs with folic acid. MTHFR SNPs are generally not considered as a problem in patients consulting for infertility, although they are a source of gestational/obstetrical complications. The negative association of MTHFR SNPs with environmental endocrine disruptors that are detected with increasing frequency in body fluids may contribute to their increased impact [10]. Gametogenesis requires methylation/epigenetic resetting, during embryogenesis in the male and at the onset of puberty in females.

This is the first large-scale epidemiologic study that addresses the relationship between MTHFR SNPs and long-standing infertility in a Caucasian male population and not restricted to female patients alone. These two SNPs should not be overlooked for patients with severe infertility of long duration, including repeat miscarriages. MTHFR SNPs in women affect oocyte quality and increase the risk of neural tube defects, miscarriages, and pregnancy complications. In considering therapeutic options, it has been clearly demonstrated that high doses of folic acid are not a reasonable option [13,14]. Supplementation with 5-methyl tetrahydrofolate (5MTHF), the compound immediately downstream from the MTHFR bottleneck, has shown real efficacy [18,32,37,42]. After the first preliminary publications [18,37], more than 300 pregnancies, mostly spontaneous, were followed by deliveries in the treated couples, with not a single report of NTD or adverse obstetrical problems. One must also consider the additional factor of slow metabolic capacity of synthetic folic acid, due to poorly efficient DHFR (dihydrofolate reductase) activity [43] upstream from the MTHFR enzyme in the folate cycle. Patients must also be informed about pleiotropic medical implications relevant to their own health, as well as to the health of future children, especially in the case of boys, as they are more sensitive to the risk of elevated homocysteine. With the consent of their parents, we have undertaken a follow-up study on homocysteine levels in children with a putative 677TT or compound heterozygous 677AC/677CT, based on parental genetic background, starting at the age of 2 years.

## Figures and Tables

**Figure 1 biomolecules-12-01086-f001:**
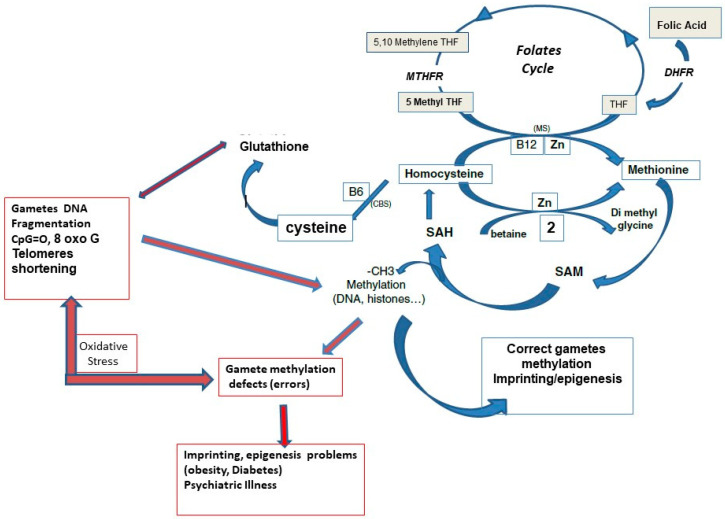
Interrelations between the one carbon cycle (methionine cycle) and the folates cycle. B6: vitamin B6, B12: vitamin B12, DHFR: dihydrofolate reductase, CBS, cystathionine beta synthase pathway, MS: methionine synthase, MTHFR: methylene tetrahydrofolate reductase, THF: tetrahydrofolate. SAH: S adenosylhomocysteine, SAM: S adenosyl methionine, CpG: cytosine phosphate guanine, Zn: zinc.

**Figure 2 biomolecules-12-01086-f002:**
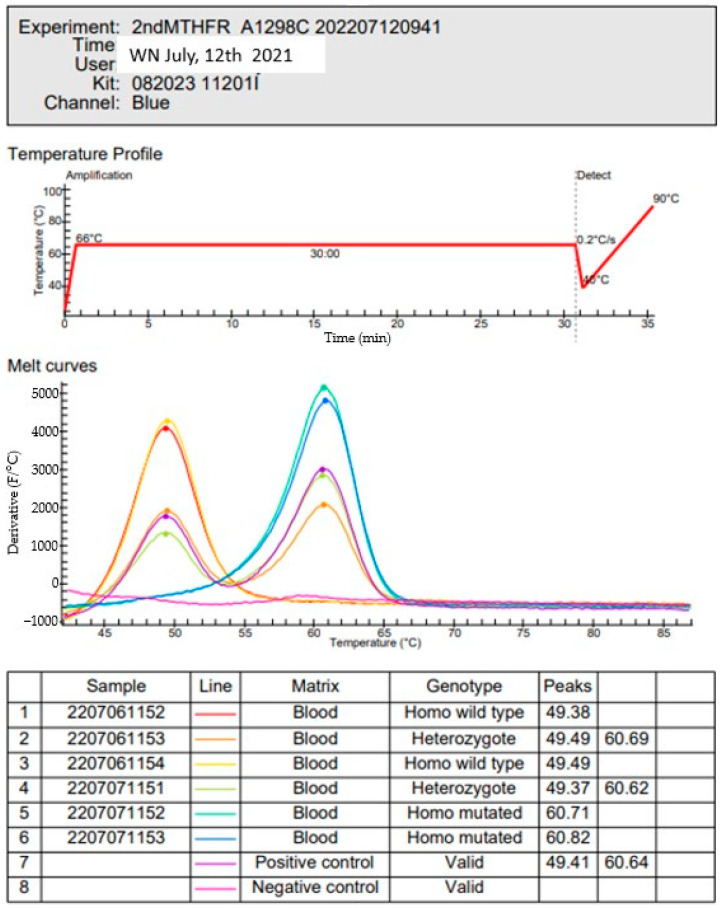
Several patients’ samples are represented in this figure: Amplification for A1298C patient 071,153. The blue line represents patient homozygous C1298C (one peak at 60.8).

**Figure 3 biomolecules-12-01086-f003:**
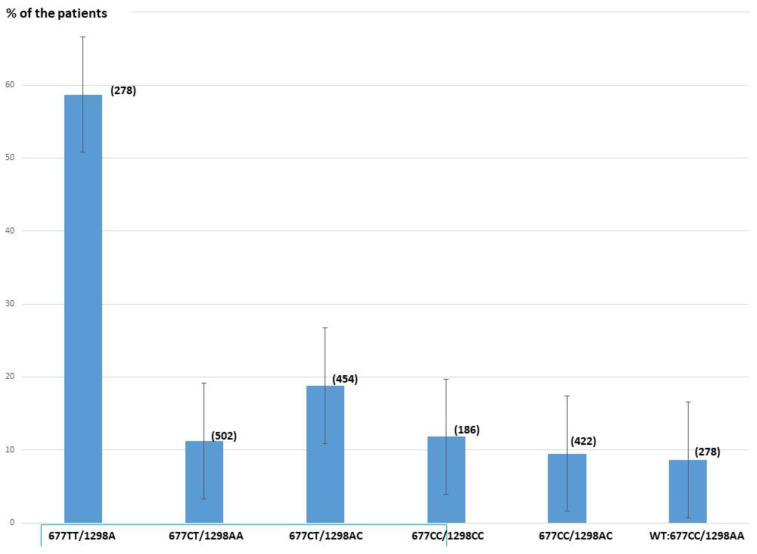
Percentage of patients with a serum Hcy level >15 micromolar for each MTHFR SNP combination. (number of patients).

**Table 1 biomolecules-12-01086-t001:** MTHFR SNP combinations in our male hypo-fertile population (percentage of the total male population).

SNP Combination	Number (% of the Population)	Number, (%) of the Patients w Hcy >15 µM)
C677C/A1298A (WT)	278(13.1%)	24 (8.6%)
T677T/A1298A	278(13.1%)	159 (57.8%)
C677T/A1298A	502(23.5%)	56 (11.2%)
C677T/A1298C	454 (21.4%)	86 (18.8%)
C677C/C1298C	186(8.7%)	22 (11.8%)
C677C/A1298C	422(19.7%)	40 (9.5%)
C677T/C1298C	4(0.2%)	-
T667T/A1298C	3 (0.2%)	1
Total	2127 (100%)	388 (18.2%)

**Table 2 biomolecules-12-01086-t002:** Odd ratios of the two main isoform combinations that significantly increase circulating homocysteine.

	Odds Ratio	Lower 95%	Upper 95%
C677T/A1298C	2.47	1.5	4
T677T/A1298A	14.14	8.7	22.9

## Data Availability

Not applicable.

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
