# Peer review of "MTHFR SNPs (Methyl Tetrahydrofolate Reductase, Single Nucleotide Polymorphisms) C677T and A1298C Prevalence and Serum Homocysteine Levels in >2100 Hypofertile Caucasian Male Patients"

_biomolecules, 2022, doi:10.3390/biom12081086_

Round 1

Reviewer 1 Report

References are missing all the way until line 58 – add them in this section of the Introduction.

Adjust font size in all of the text according to manuscript guidelines.

In the introduction section, better emphasis with more details should be added on the gap in the literature this study is filling and exact novelty to the field.

Where did this study take place?

I would remove Figure 1 from the Methods, and if not deleted, move it to Introduction somewhere more appropriately.

Was this study approved by the institutional Ethics Committee?

Mathods section is extremly limited and insufficent... especially part about subjects.

Remove referencing to Table 1 from the chapters’ sub-captions and put it into text.

In Discussion, more emphasis should be put into clinical implications derived from the results of this investigation. Discussion section is very limited and short...

Add limitations of the study in the Discussion.

Better conclusion that is connected directly to the results from this investigation should be incorporated.

Author Response

ANswers to reviewers comments.

Rev 1 :

We have published several papers, case reports in JARG (Servy ey al, Jacquesson Fournols et al. Goyco Ortiz et al..). We have recently published a paper in J Women Health on Endometriosis patients. The 2 most striking impact is on Repeated IVF ICSI failure and on repeat miscarriages (see Servy et al. “In the group of 14 patients having undergone 61 miscarriages (4.36 per patients), we recorded 4 term deliveries and 8 ongoing pregnancies (> 3 months) the overall ongoing pregnancy rate is 86.7%” In this group of patients we have now 15 deliveries. One of the most interesting cases is in the Jacquessson Fournols paper: a strict paternal effect. The impact on multiple ART failures patients is also interesting. We collect now most of the French patients who do not want to have oocyte donation. We will make a full synthetic paper end 2023, 5 yrs after our experience with the 2 isoforms testing. We do think that the combo heterozygous cases are too important to have only the 677TT

We have the data on 2400+ women. Some have been introduced in the discussion. But the paper is focused on the male aspects obviously often neglected even if literature is now full of papers linking methylation defects/oxidative stress/infertility

All the forms for the consents have been sent to the editor. AS mentioned, Genetic testing is highly controlled, building a control group is forbidden by law. MTHFR testing is now a classical parameter but not a first row one. The physicians need to have agreements. If the consent is not signed by the 3 parties: clinician, patient and laboratory, the test cannot be made

The figure describing the curves has been added.

The 2 “fertile groups” tested for MTHFR, described in literature have been added. Some papers (Cirillo et al) do not test or mention the combo form. So impossible to quote them on the basis of our results. Rather high prevalence around 20% and the impact of Homocysteine. One has to understand the impact on the embryo for these patients can be important, if the 2 members of the couple are Combo.  Our population is a special hypoferile population

The text has been rereviewed by Kay Elder, Bourn Hall clinic, Cambridge

10 refs have been added. The forms of quotations in the text have been modified

Rev2:

References have been introduced earlier in the introduction.  Refs have been added.

The intervention of the ethic committee not needed; the consent form has been sent to the editor. Genetic testing is severely controlled in France

We have added some refs on the paternal effects. They are rather rare as “the MTHFR blame was rather put on the women”.  It is obvious, from the data of Enciso et al; (quoted here) that in the embryo, the 2 gametes have their (^*positive or negative role). The other point is that the prevalence of the Combo (double Heterozygoty) is rather high: the impact on Hcy and on the embryo is not benign

The studies have been performed in clinics having the ARS agreement in Paris. The Laboratory has agreement for Genetic and cytogenetics as required by law; The clinician have also an agreement for prescription. The consent form for the test has been sent to the editor

The discussion has been modified; we have added some data concerning the woman side; basically, the study is rather focused on men

Reviewer 2 Report

The article is interesting because it is related to testing genetic polymorphisms which are usually performed in female individuals with low fertility rates and recurrent abortions. The article emphasizes the need of testing the males of infertile couples as well.  The results are interesting with a high ODDs ratio, but some information is missing, how many females were homozygotes? how many of the couples had successful pregnancies? Did the patients describe any comorbidities? Did the fertility increase with the oral intake of methylfolate? 

The other issue which is confusing is ethics committee approval. The text includes several issues and in the end, the study did not require ethical approval. Do the authors mean that written consent was taken before the test and the ethical committee does not require further action? There should be a statement from the ethical committee answering the query. 

In addition, include in supplementary figures, as done in the published manuscript in french https://doi.org/10.1016/j.gofs.2020.02.015  the curves of the tests.SNP 

The discussion and conclusion should be modified. The incidence of SNP is very high and should be compared with more studies. Furthermore, the study has limitations, the lack of controls from a similar population for example.

Finally, the article needs a good review in the English language, several parts are difficult to understand. For example, We observed that the genetic status with highest 24 prevalence is the heterozygous C677T, followed by the combined Heterozygous C677T/A1298C, 25 and then A1298C: these three variants represent 65% of our population

Please refer to these data as haplotype analysis accordingly. 

Author Response

(The authors gave the same response as above.)

Round 2

Reviewer 1 Report

The authors have improved only a few selected aspects of initial review. The MS is still significantly below the publishing standards.

Reviewer 2 Report

The changes introduced by the authors are valuable. The article now it is improved. It is suitable forpublication